# Dietary Impacts on Changes in Diversity and Abundance of the Murine Microbiome during Progression and Treatment of Cancer

**DOI:** 10.3390/nu15030724

**Published:** 2023-01-31

**Authors:** Holly Paden, Nikola Kurbatfinski, Jelmer W. Poelstra, Kate Ormiston, Tonya Orchard, Sanja Ilic

**Affiliations:** 1Department of Human Sciences, Ohio State University, Columbus, OH 43210, USA; 2Molecular and Cellular Imaging Center (MCIC), Ohio State University, Wooster, OH 44691, USA

**Keywords:** cancer, murine microbiome, alpha diversity, dietary composition, chemotherapy, disease progression, omega-3, sucrose

## Abstract

The intestinal microbial population is recognized for its impact on cancer treatment outcomes. Little research has reported microbiome changes during cancer progression or the interplay of disease progression, dietary sugar/fat intake, and the microbiome through surgery and chemotherapy. In this study, the murine gut microbiome was used as a model system, and changes in microbiome diversity, richness, and evenness over the progression of the cancer and treatment were analyzed. Mice were categorized into four diet cohorts, combinations of either high or low sucrose and high or low omega-3 fatty acids, and two treatment cohorts, saline vehicle or chemotherapy, for a total of eight groups. Fecal samples were collected at specific timepoints to assess changes due to diet implementation, onset of cancer, lumpectomy, and chemotherapy. *Akkermansia muciniphila* abundance was very high in some samples and negatively correlated with overall Amplicon Sequence Variant (ASV) richness (r(64) = −0.55, *p* = 3 × 10^−8^). Throughout the disease progression, ASV richness significantly decreased and was impacted by diet and treatment. Alpha-diversity and differential microbial abundance were significantly affected by disease progression, diet, treatment, and their interactions. These findings help establish a baseline for understanding how cancer progression, dietary macronutrients, and specific treatments impact the murine microbiome, which may influence outcomes.

## 1. Introduction

The prevalence of cancer remains high [1]. Currently, it is expected that there will be 1,918,030 incident cases of cancer and 609,360 cancer deaths, this year [1]. Breast cancer is considered “fairly common” by the National Institute of Health [2], and is anticipated to account for 15% (287,850) of those new cases and 7.1% (43,250) of cancer deaths [1]. Women have a 13% lifetime chance of developing breast cancer, requiring them to receive therapeutic treatment [2].

An important aspect of achieving the best cancer treatment outcomes is maintaining patient health against infectious disease and opportunistic pathogens throughout treatment. When patients receive any systemic treatment against breast cancer (e.g., chemotherapy or chemo-radiation), a surgery, or a combination of treatments, their immune system becomes depressed [3]. These therapies target rapidly dividing cells, such as aggressive cancer cells, and many cells which are vital to host immunity (red blood cells, lymphocytes, macrophages, and neutrophils) [4], leading to patients’ increased susceptibility to infections [5] including, among others, foodborne pathogens such as *Salmonella* spp., STEC (Shiga-toxin producing *E. coli*), and *Listeria monocytogenes* [5,6]. As both the treatment and disease progress, maintaining gastrointestinal health becomes increasingly difficult [7]. In particular, the intestinal microbial population may be affected, leading to dysbiosis, and triggering additional inflammatory responses.

Both the composition of the intestinal microbial populations and their diversity are vital to maintaining patient health [8]. There is a cyclical relationship between the gut microbiome and cancer treatment responsiveness, in which each impacts the other [9]. It has been well documented that the state of the gastrointestinal (GI) microbial community correlates with efficacy and outcomes related to chemotherapy [10,11]. The diversity and composition of the gut microbiome is posited to affect the outcome of cancer treatment [12,13], and a depleted microbiome with minimal bacterial diversity is believed to further depress immune system efficacy. This increases the susceptibility of the host to infection, and colonization of the GI tract with unfavorable bacteria has been shown to worsen disease outcomes and can cause long-term damage to the host immune system [14,15].

Diet is inextricably linked to shaping the microbiome [14], as well as correlated to the onset of cancer and its treatment outcomes. Diet also directly impacts inflammation and can indirectly affect it through changes in the microbiome [16,17,18] For instance, it has been shown that a diet high in fat and red meat correlates to an elevated risk of colorectal cancer diagnosis [19]. Diet has been particularly highlighted as a factor which affects the diversity and composition of the GI microbiome [20,21]. Both the diversity and population composition of the microbiome are important factors when considering host resilience [22]. Decreased diversity has been associated with a high fat, high sugar diet, a combination that is typically considered a “Western” diet [22]. This kind of diet has been linked to inflammation and dysbiosis through the microbiome [16]. For instance, a diet high in fats and sugars is also associated with a predomination by Mollicutes, a parasitic bacterial class in the Firmicutes phylum that is commonly linked to inflammation [23,24].

However, little research currently exists on the effect of cancer treatments on the state of the microbiome, and the factors that mediate that relationship. This is vital to understand, as the toxicity of chemotherapy may reduce the diversity of the microbiome, thus increasing GI side-effects of treatment and increasing patient susceptibility to opportunistic infections. Because dysbiosis of the microbiome may lead to patient health concerns [14,25] and, ultimately, cause disruptions in patient treatment or reduce treatment options for those patients, it is critical to fill this gap in knowledge. Until the impact of cancer progression and the intersection of diet and treatment on the microbiome is better understood, it will be challenging to appropriately treat cancer patients and to mitigate potential side-effects.

The objective of this study is to determine changes in the richness and diversity of the gut microbiome over the course of cancer and subsequent treatment, dependent on different dietary patterns. This study uses fecal samples as a proxy for analysis, and tracks changes in the microbiome through breast cancer progression and therapy in mice fed diets with various levels of fat and sugar, to emulate dietary recommendations [26] and common intake levels in a Western diet [27]. In this study, we follow a timeline of breast cancer onset and progression, to surgical treatment, to chemotherapy. This timeline is designed to represent a typical experience of cancer patients. We hypothesize that this progression will result in (i) a decrease in the diversity of the gut microbiome, independent of diet, and (ii) an increased presence of pathogenic commensals and bacterial species commonly linked to inflammation. Additionally, we hypothesize that consumption of a high omega-3 or high sucrose diet will exacerbate negative microbiome effects, with a combination diet compounding this impact. Finally, treatment with chemotherapy is expected to further decrease the diversity and density of the gut microbiome population, whereas a lack of chemotherapeutic treatment is not expected to negatively impact the microbiome.

## 2. Materials and Methods

### 2.1. Animals

Mice were chosen as the model organism for control of diet, control of cancer, control of treatment, cost of maintenance, and space availability. A total of 116 C57bl/6 female mice, 7–8 weeks old, were purchased from Charles River Laboratories (Wilmington, MA, USA). After arrival, mice were housed in an Association for Assessment and Accreditation of Laboratory Animal Care-approved vivarium and fed a normal chow diet, prior to the start of this experiment.

### 2.2. Diet

Mice were randomly assigned to one of four diet groups (*n* = 29/group) using a lottery system, and each group was assigned to a semi-purified diet produced by Research Diets, Inc. (Newark, NJ, USA) beginning one week after recovery from ovariectomy surgery. The first diet was a low sucrose (9% of kcal), high omega-3 fatty acid diet (LS, HF), with 2% of daily kcal coming from omega-3 fatty acids. Microencapsulated EPA and DHA was present in this diet at a ratio of 1.5:1. High omega-3 consumption is expected to be associated with decreased inflammation [28]. The second was a low sucrose, low omega-3 fatty acid diet (LS, LF) with alpha linolenic acid (plant-based) at a low dose to prevent essential fatty acid deficiency; EPA and DHA were not components of the diet. Low omega-3 diets are representative of common fatty acid intakes in Western culture, in which omega-3 fatty acids are typically from plant sources [27]. Low sucrose diets were constructed in accordance with dietary recommendations [26]. Third was a high sucrose (47% of kcal), high omega-3 fatty acid diet (HS, HF). High sucrose has been documented to counter the positive effects of high omega-3 fatty acid consumption [29]. Fourth was a high sucrose, low omega-3 fatty acid diet (HS, LF). The combination of low omega-3 intake compounded with high sucrose makes this diet the closest representative of a “Western diet”. All semi-purified combinations had food dye added, with colors of red (LS, HF), yellow (LS, LF), blue (HS, HF), and green (HS, LF). Diets were stored under refrigeration and changed every three to four days in cages to prevent fatty acid oxidation.

All diets used in the experiment were comparable or equivalent in calories, minerals, vitamins, and macronutrient distribution. Diet composition and ingredients were previously described in Ormiston et al., 2021 [27]. Additional data of mouse body weights, tumor weights, and proportional tumor weight at necropsy, by noted time point, were collected by the Orchard laboratory throughout the project.

### 2.3. Experimental Design

Mice were acclimated to facilities for approximately two weeks, then underwent an ovariectomy to mimic the postmenopausal state common to the majority of breast cancer patients [30]. Following ovariectomy, mice were weighed and housed individually. After recovering from ovariectomy surgery for one week, the semi-purified diets were introduced (Appendix A).

One week following initiation of semi-purified diets, all mice were inoculated with 100 µL of 1 × 10^5^ E0771 murine mammary cancer cells into right abdominal (4th) mammary fat pad. Twelve days after inoculation with cancer cells, all mice underwent lumpectomy in which the intact tumor was excised along with any visible fat pad. Eight days following the lumpectomy procedure, body weight assessment was performed again. Ten days following lumpectomy, mice from each dietary cohort were treated with either a saline vehicle injection or a chemotherapy injection (*n* = 16 per chemo/diet group and *n* = 13 per vehicle/diet group). Chemotherapy composition was 9 mg/kg body weight of doxorubicin and 90 mg/kg body weight of cyclophosphamide (50% of human equivalent dose). The experiment ended early, one week after either saline vehicle or chemotherapy injection, because tumor regrowth was detected. All mice were euthanized, according to IACUC guidelines. Briefly, mice were injected with euthasol (270 mg/kg) and when deeply anesthetized, cardiac perfusion was performed, and tissues were collected. Tumors and organs were weighed and stored in −80 °C until further analysis.

### 2.4. Sample Collection

Throughout the course of the study, tumors were palpated, to assess growth, and were measured using calipers. During the experiment, mouse feces were collected, and samples were stored in pre-labeled collection tubes. Each mouse had its own cage, and a sterile pair of forceps was used for each collection. Up to 2 mL of feces was collected for each mouse on days D-1 (baseline), D6 (diet effect), D16 (cancer effect), D29 (surgery effect), and D35 (treatment effect). Bedding was changed after each fecal collection and three days following administration of chemotherapy/vehicle injections. Individual day/mouse tubes were stored in labeled and sealed Whirl-Pak bags and preserved at −80 °C.

### 2.5. DNA Isolation and Processing

Using a random number generator, four representative samples were selected for each collection timepoint, diet, and treatment combination, with the exception of samples collected one day before diet introduction, for which six samples were collected. Selected fecal samples were processed according to instructions given for the QIAmp^®^ PowerFecal^®^ DNA isolation kit (Qiagen; Hilden, Germany, 2017). Upon completion of DNA isolation, we measured the concentration and purity of each sample using NanoDrop (Thermo Scientific Nanodrop 1000 Spectrophotometer). Each sample was then diluted to a concentration of 5 ng/µL and suspended in Tris buffer solution (ThermoFisher Scientific; Waltham, MA, USA). Following isolation, samples were sent to the Molecular and Cellular Imaging Center (MCIC) core facility at the Ohio State University, Wooster campus, for 16S ribosomal genetic profiling. A genomic library was generated after amplification with the following primers, as in previous studies [31,32]: Forward-S-D-Bact-0564-a-S-15 (5′-AYT GGG YDT AAA GNG) and Reverse-S-D-Bact-0785-b-A-18 (5′-TAC NVG GGT ATC TAA TCC), amplifying a fragment of 206 bp.

### 2.6. Morphological Analysis

Diet and treatment effects on morphological measurements of mice were analyzed over the course of the experiment, using a regular Analysis of Variance (ANOVA) for body weight and body weight change, and an ANOVA using M-estimators (pbad2way() function from the R package WRS2, version 1.1.3 [33]) for the non-normally distributed measurements of tumor weight and tumor weight as a proportion of body weight.

### 2.7. Microbiome Data Analysis

All code used for the analyses can be found in our GitHub repository for this study (https://github.com/jelmerp/mouse-cancer-metabarcoding, accessed on 26 January 2023). Computation was performed at the Ohio Supercomputer Center [34] using project PAS0471. After confirming that FASTQ files for all samples were of an appropriate quality for analysis using FastQC (version 0.11.8 [35]) and MultiQC (version 1.11 [36]), primers were removed using cutadapt (version 3.4 [37]).

All downstream analyses were performed in R (version 4.1.1 [38,39]). The R/Bioconductor package *DADA2* (version 1.16 [40]) was used to generate a count table with counts of each inferred Amplicon Sequence Variant (ASV) for each sample. Briefly, this consisted of the following consecutive steps: sequence quality filtering and trimming (filterAndTrim()function), dereplication (derepFastq() function), sequence error modeling (learnErrors() function), denoising/ASV inference (dada() function), merging forward and reverse read pairs (mergePairs() function), creating a sequence table (makeSequenceTable() function), inferring and removing chimeric ASVs (removeBimeraDenovo() function), and taxonomic assignment (assignTaxonomy() and addSpecies() functions) using the Silva [41] database version 138.1 (available at https://zenodo.org/record/4587955, accessed on 26 October 2021) (Appendix A). Next, several filtering steps were performed: ASV were filtered by length (retaining only ASVs that were 204–209 bp long), contaminants were inferred and removed with the R/Bioconductor package *decontam* (version 1.14.0 [42]) using a negative control and DNA concentration measurements, ASVs assigned to the order Chloroplast, the family Mitochondria, or the domain Eukaryota were removed, and samples with a total ASV count below 1000 were removed.

A phylogenetic tree for all ASVs was inferred using the R package *phangorn* (version 2.8.1 [43]). The R/Bioconductor package *phyloseq* (version 1.38.0 [44]) was used to store the resulting count table, taxonomy table, phylogenetic tree, and a metadata table as a single R object and perform a number of downstream analyses, such as calculating weighted UniFrac distances among samples, performing a Principal Coordinate Analysis (based on UniFrac distances), and agglomerating counts at higher taxonomic levels. To examine overall differences in microbiome composition between days, diet, and treatment, we used the adonis2() function from the R package *vegan* (version 2.5.7 [45]) to run a PERMANOVA with the weighted UniFrac distance between samples as the responding variable. Pairwise PERMANOVA comparisons were done with the *pairwise.adonis2* function from the R package *pairwiseAdonis* (https://github.com/pmartinezarbizu/pairwiseAdonis, accessed on 28 December 2022 version 0.4.1).

In the Supplementary, sequence data removal and filtering in different steps of the pipeline are visualized in Appendix A and taxonomic assignment success for ASVs is visualized in Appendix A. An average of 158,455 reads per sample (minimum: 54,644; maximum: 301,865) were sequenced, and an average of 142,898 (minimum: 47,581; maximum: 284,075) were retained after quality control measures at the end of the ASV inference pipeline. Taxonomy was assigned to ASVs at high rates at the order level (97.48%), declining at the family (81.68%), genus (55.66%), and especially species (5.59%) levels.

Taxonomic richness was estimated for each sample and compared between days, diet, and treatment using the R package *breakaway* (version 4.7.6 [46,47]). Other measures of alpha-diversity (Simpson and Shannon indices) were estimated and compared among days, diet, and treatment using the R package *DivNet* version 0.4.0 [48]). The approaches implemented in these packages robustly estimate diversity measures while accounting for unobserved species and not requiring rarefaction [49].

Differential abundance analysis at the ASV, genus, and family level was performed using the R/Bioconductor package *ALDEx2* (version 1.24.0 [50]). Reported *p*-values were adjusted using the Benjamini–Hochberg multiple-testing correction method. For consistency with this analysis, ASV counts of differentially abundant taxa were visualized after normalization with the *ALDEx2* function aldex.clr().

A list of taxa of interest (*n* = 43) was compiled, based on previously reported commensal human bacteria which are associated with influencing cancer and treatment outcomes or are opportunistic commensal pathogens associated with foodborne disease. Representatives of these taxa or the closest higher taxonomic level that was present in our data were singled out to assess differential abundance, plot abundances, and, where appropriate, perform separate ordination analysis.

## 3. Results

### 3.1. Morphometric Measures 

Body weight of adult mice significantly increased over the 5 weeks of the experiment (F(2, 64) = [1.103 (0.367)], *p* < 0.01) regardless of diet (baseline: 19.434 g ± 2.399 g; necropsy: 21.736 g ± 1.755 g; d = 1.096, r = 0.480) (Figure 1a).

Food intake was significantly higher in both diet groups of mice consuming high sucrose, compared to mice consuming low sucrose. Tukey’s test for multiple comparisons showed that the mean food intake for LS, LF was significantly lower than either HS, HF or HS, LF (*p* = 0.093, 95% CI = (−0.273 to −0.010); *p* = 0.0253, 95% CI = (−0.278 to −0.013), respectively). The same test showed that mean food intake of LS, HF was significantly lower than LS, LF; HS, HF; and HS, LF (*p* = 0.024, 95% CI −0.274 to −0.014; *p* < 0.0001, 95% CI = −0.416 to −0.156; *p* < 0.0001, 95% CI = −0.421 to −0.159, respectively). The mice that received chemotherapy showed significantly less weight gain compared to the control group (F(2, 64) = [−1.856 (0.522)], *p* < 0.001). In a two-tailed *t*-test, body weights were significantly higher in the vehicle treatment group than in the chemotherapy treatment group, by the end of the study (*t*(107) = 4.179, *p* < 0.0001) (Figure 1b).

### 3.2. Microbial Diversity, Richness, and Abundance

Overall, microbial diversity significantly changed over the course of the experiment (i.e., by day) and differed by diet. The model for day and diet was found to be the best fit; effects of day, diet, and their interactions on richness are shown in Table 1. With LSHF acting as the intercept, significant differences were found between the intercept and LSLF (F(2, 65) = [−36.3 (14.1)], *p* < 0.01), as well as HSHF (F(2, 65) = [−90.5 (13.3)], *p* < 0.001). Richness significantly decreased six days after diet implementation (F(2, 65) = [−132.4 (15.7)], *p* < 0.001). The trend of decreasing richness continued with day 16 after diet implementation (F(2, 65) = [−164.3 (16.1)], *p* < 0.001), day 29 (F(2, 65) = [−219.1 (16.2)], *p* < 0.001), and day 35 (F(2, 65) = [−263.3 (11.6)], *p* < 0.001) (Figure 2). Effects of diet, the interactions between day and diet, and the interactions between diet and treatment are shown in the Appendix A.

Other alpha-diversity measures, such as the Shannon index, also take into account evenness in abundance among ASVs. The Shannon index was also significantly affected by day and diet (Figure 3), as well as by treatment. Day-only and treatment–diet interaction reporting can be found in the Appendix A.

### 3.3. Overall Microbiome Composition

A PERMANOVA analysis by day and diet, using UniFrac distance as the respondent variable, showed significant differences in overall microbiome composition by day (F(3, 76) = 7.0097, R^2^ = 0.2075, *p* = 0.0001), diet (F(3, 76) = 2.1859, R^2^ = 0.0647, *p* = 0.0001), but not by the interaction between the two (F(9, 70) = 1.087), R^2^ = 0.0965, *p* = 0.32457). Pairwise comparisons among diets in the above model revealed significant differences between the two low sucrose diets (R^2^ = 0.0707, *p* = 0.006), and between LSLF and HSHF (R^2^ = 0.04921, *p* = 0.024). When considering the chemotherapy treatment in a model that also included diet, significant differences were observed between mice that had received chemotherapy versus placebo (F(1, 30) = [2.7814], R^2^ = 0.0974, *p* = 0.04170). Ordination plots using principal coordinate analysis indicated that one ASV, which was classified as *Akkermansia muciniphila*, had a large impact on this analysis (Appendix A). This was overall by far the most abundant ASV (ASV1 mean proportional abundance = 0.233; 26 of 86 samples showed ASV1 proportional abundance > 0.5), but its pattern of abundance among samples was clearly bimodal after baseline (Figure 4). Of individual mice that had been randomly sampled at more than one timepoint, 11 experienced an increase of *A. muciniphila* abundance, while four experienced a decrease, and two experienced both an increase and a decrease at separate time points (Appendix A). The abundance of *A. muciniphila* and overall sample taxonomic richness was negatively correlated (r(64) = −0.55, *p* = 3.00 × 10^−8^).

### 3.4. Differential Abundance

An analysis of differential abundance showed that a total of 65 ASVs significantly differed in abundance by day, as did 13 genera and 11 families (Figure 5).

In a model that considered both the day of disease progression and the diet, one ASV showed a significant increase by day (ASV 21; *p* = 4.04 × 10^−4^). ASV50 was identified as *Oxalobacter formegenes*, and was significantly less abundant in mice fed low omega-3 diets, with low or high sucrose supplementation, respectively (*p*1 = 6.41 × 10^−3^, *p*2 = 2.86 × 10^−2^) (Figure 6A). This relationship was retained in the model that considered chemotherapy treatment and diet interactions (ASV 50; *p*1 = 7.88 × 10^−3^, *p*2 = 2.52 × 10^−3^, low omega-3 diet with low and high sucrose, respectively). The genus *Romboutsia* increased by disease progression day in the day–diet interaction model (*p* = 4.12 × 10^−2^). At the family level, *Peptostreptococcaceae* increased by disease progression day (*p* = 3.86 × 10^−2^) (Figure 6B) and *Oxalobacteraceae* showed lower abundance in mice consuming either of the low-fat diets with low and high sucrose, respectively (*p*1 = 4.92 × 10^−4^, *p*2 = 2.67 × 10^−4^).

When treatment and diet were considered, the family *Streptococcaceae* was less abundant in mice that received chemotherapy (*p* = 4.00 × 10^−2^) and *Oxalobacteraceae* were significantly lower in the low sucrose, low omega-3 diet group, regardless of treatment (*p* = 3.66 × 10^−2^).

### 3.5. Taxa of Interest

Genera identified in the samples included *Bacteroides*, *Bifidobacterium*, *Enterococcus*, *Lactobacillus*, *Legionella*, *Mycobacterium*, *Proteus*, *Pseudomonas*, and *Streptococcus*. *Bilophila wadsworthia*, *Staphylococcus aureus*, *Klebsiella* spp., and *Peptococcus* spp. were identified to the genus and species level. These four taxa did not show significant differences in abundance by disease progression, diet, or treatment.

*Bacteroides intestinalis* (*p* = 1.72 × 10^−7^), *Bacteroides acidifaciens* (*p* = 3.06 × 10^−2^), *Bacteroides uniformis* (*p* = 1.88 × 10^−5^), and the genus as a whole (*p* = 1.81 × 10^−4^) increased during the progression of cancer, as did the genera *Lactobacillus* (*p* = 3.47 × 10^−4^) and *Streptococcus* (*p* = 3.93 × 10^−2^).

The identified families included *Peptostreptococcaceae*, *Clostridiaceae*, *Enterobacteriaceae*, *Oscillospiraceae*, and *Listeriaceae*. Three ASVs from the genus *Romboutsia*¸ in the family *Peptostreptococcaceae*, increased by day, as did the overall family (*p* = 8.58 × 10^−13^). Additionally, the abundance of the *Peptostreptococcaceae* family was impacted by the interaction between day and diet (*p* = 3.86 × 10^−2^). The *Clostridiaceae* family fluctuated by day (*p* = 3.77 × 10^−2^).

At the order level, Campylobacterales and Bacteroidales were detected and Bacteroidales abundance was significantly impacted by day (9 ASVs). At the class level, Gammaproteobacteria (189 ASVs) was impacted by day and diet.

## 4. Discussion

In this study, we have demonstrated changes in diversity and composition of the mouse microbiome during cancer progression and treatment. As anticipated, alpha-diversity decreased as the disease progressed and differed between diet types. However, though there were significant differences between the diets, conclusive results could not be drawn on if certain nutrients protected or exacerbated this decrease in diversity. It is well documented that diet is a major determinant of the richness and diversity of the microbiome [51], as supported by our results. Oligosaccharides are known to be used by opportunistic pathogens within the gut, and can lead to, or worsen, the state of disease and limit treatment opportunities [52]. Though no pathogenic commensals were able to be identified to the species level, there were multiple ASVs, families, and classes of focal taxa which were impacted by dietary patterns, which warrants further investigation.

Diets high in sugar or fat have been previously shown to promote inflammation associated with the gut microbiome [16]. Sucrose, in particular, has been associated with changes in the microbiome and to permeability of the intestinal mucosal barrier, an important factor in maintenance of host immunity [53]. This study used a high sucrose, low omega-3 diet to simulate common Western dietary patterns, as the Western diet typically involves obtaining fatty acids from plants, rather than marine sources [27]. While some dietary effects were seen, more investigation is required, as dietary intake and its effects on the microbiome may be much more intricately linked to additional factors, such as genetics and environment. The changes in abundance, from this study, can largely be attributed to a decrease in overall richness throughout the disease progression, rather than explicitly to diet.

The abundance of several ASVs, genera, and families were significantly impacted by diet, treatment, disease progression, and their interactions. However, a distinct link was not able to be determined between disease progression or diet and an increase in pathogenic commensals or inflammatory bacterial species, as hypothesized. Unexpectedly, *Akkermansia muciniphila* demonstrated a bimodal polarization after baseline, without regard for diet. *Akkermansia* is a mucin-degrading bacterium [54] and its presence is highly associated with favorable outcomes for cancer patients [55]. It is usually a minor resident of the gut, and is the only known bacterium in the *Verrucomicrobia* phylum [14]. It is generally associated with positive health outcomes and increased longevity [8], and its presence in the murine gastrointestinal microbiome has previously been shown to increase in mice fed fish-oil [17]. *Akkermansia* populations have been shown to be negatively impacted, in mouse model systems, by the administration of cyclophosphamide [10]. In this study, no association was found between *Akkermansia* and diet or treatment. The abundance distribution was bistable and switched randomly, in direction and magnitude, within individual mice, for reasons unable to be explained by the scope of this experimental design. Further analysis is needed to determine why this occurred.

The microbiome influences innate immunity [56] and affects the response of the host to cancer therapies, such as chemotherapy or immunotherapy [57]. A cyclical relationship exists between the composition and diversity of the microbiome and cancer therapy [58]. In this study, we specifically investigated the combination of doxorubicin and cyclophosphamide and found significant changes in richness and alpha-diversity associated with chemotherapy treatment, dependent on diet. Some microbial communities contribute to resilience against doxorubicin toxicity [59], but the administration of doxorubicin still causes changes in the composition of the gut microbiome [60]. Cyclophosphamide has been shown to drive the murine microbiome towards dysbiosis, significantly decreasing *Lactobacilli* and *Enterococci* populations, while increasing *Clostridium* group IV [11]. Depletion of the diversity of the intestinal microbial population is associated with negative cancer outcomes, while in a healthy individual, the composition of the microbiome remains stable for 60% of bacterial species and diversity is commonly high [61]. Many commensals, such as *Bifidobacteria*, have been shown to be susceptible to the toxicity of chemotherapeutic treatments and can become depleted [62], allowing for an overabundance of opportunistic pathogens. Despite the clear risk that this combination may pose to the host microbial population, combinations of anthracyclines with cyclophosphamide have been the primary chemotherapeutic treatment for metastatic breast cancer since the 1990s [63]. Yet, there are still relatively few studies investigating the interaction between chemotherapy and the gut microbiome [64].

Beyond this analysis of the interplay between cancer treatment and the gastrointestinal microbiome, there are also very few studies that assess the relationship between the microbiome and cancer itself. In one study, cancer progression was associated with a decreased bacterial load in the microbiome: patients with stage 1 cancer showed a greater total bacterial DNA load than patients with stage 2 or stage 3 cancer [25]. However, these results were analyzed from the perspective of the impact of the microbiome on tumorigenesis, rather than assessing the interaction between cancer progression and the microbial population [65]. In this study, we found that microbiome richness significantly decreased after onset of cancer, as did alpha-diversity. Future studies should consider that onset of cancer correlates with a decrease in microbiome diversity, which may further exacerbate tumor progression. Additionally, this study was performed only with breast cancer cells, and so results should not be extrapolated to different types and sites of cancer.

Despite general profiles being documented, interactions between the gut microbiota and the host are specific to the host, and are dependent on genetic, historical, and physical factors, making the composition and cross-talk of the microbiome unique [66]. Therefore, there is no one specific “optimal” microbiome population, as there are many factors which shape the microbial population specifically for the host [67], and it is commonly accepted that a healthy microbiome is one in which the pathogenic bacterial populations are in balance with the symbiotic populations [68].

In this study, the richness of the gastrointestinal microbiome began decreasing prior to inoculation with breast cancer cells, and the presence of *Akkermansia muciniphila* occurred independently of diet and prior to cancer inoculation. These results may be linked to the ovariectomy performed on the mice prior to sample collection and diet administration. Previous studies have indicated that surgery allows for bacterial translocation, and this is associated with changes in the composition of the gastrointestinal microbiome [69]. Therefore, the changes in richness observed in this study may also be attributable to the performance of surgery prior to sample collection.

Mice are effective preclinical models and help to establish an understanding of the mechanism of changes in the microbiome due to treatment [51]. However, this model is limited by differences in anatomy and genetics, and so is not a perfect comparator for humans [66]. This limitation can be addressed in future studies by using other model systems and, eventually, human patients in order to more accurately understand relevant microbiome shifts throughout the onset and progression of cancer and treatment. Additionally, fecal samples have been accepted as an adequate proxy for assessing the microbial community of the gut microbiome and are less invasive than taking samples directly from the GI tract. However, community structure is frequently lost, leading to an incomplete picture of the microbiome [70]. Future studies would benefit from taking samples directly from the GI tract, when possible.

## 5. Conclusions

In this study we determined how richness, evenness, and composition of the murine gastrointestinal microbiome are impacted by the progression of cancer, macronutrient dietary composition, chemotherapeutic treatment, and the interplay between the three. The microbiome has been well analyzed for its contribution to cancer treatment outcomes, but little focus has yet been given to how it is cyclically impacted by cancer, even prior to treatment. Projections for new cancer cases remain high, which means that, at any given time, there are a large number of patients receiving treatment. As the composition and diversity of the microbiome contributes to host immunity and resilience, understanding the complex relationship between disease progression and microbial populations is vital to improving treatment outcomes. These results should be considered, when designing clinical studies, with regards to how the richness and diversity are negatively impacted by changes in diet and the progression of cancer. The overgrowth of commensal pathogens and the depletion of beneficial commensals needs to be quantified and addressed. This study contributes to a better understanding of how the microbial population is impacted by cancer treatment and the course of the disease, leading to potentially better patient support and improved treatment outcomes.

## Figures and Tables

**Figure 1 nutrients-15-00724-f001:**
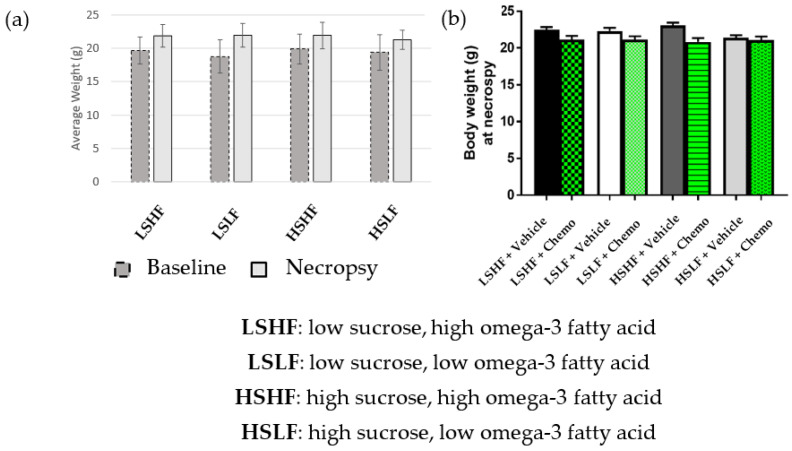
(**a**) Body weight of mouse diet cohorts at diet implementation and necropsy by diet only; (**b**) necropsy body weight of mouse diet cohorts by chemotherapy or saline vehicle treatment.

**Figure 2 nutrients-15-00724-f002:**
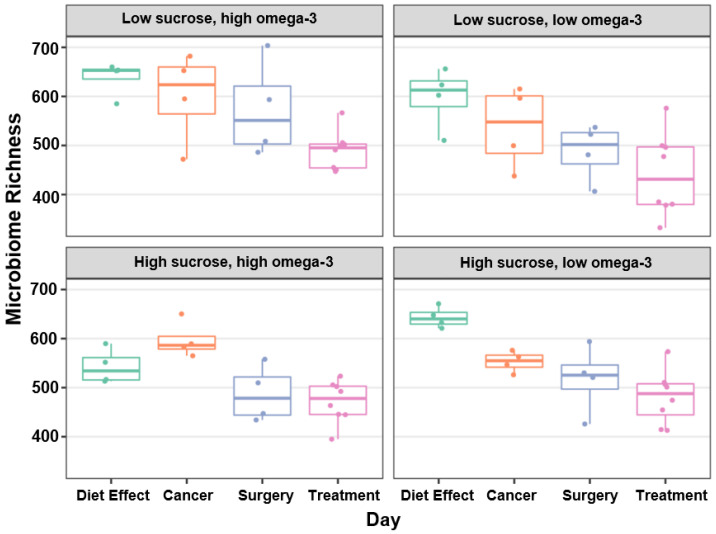
Estimated richness over progression of cancer and treatment, by diet. Microbiome population richness significantly decreased over time.

**Figure 3 nutrients-15-00724-f003:**
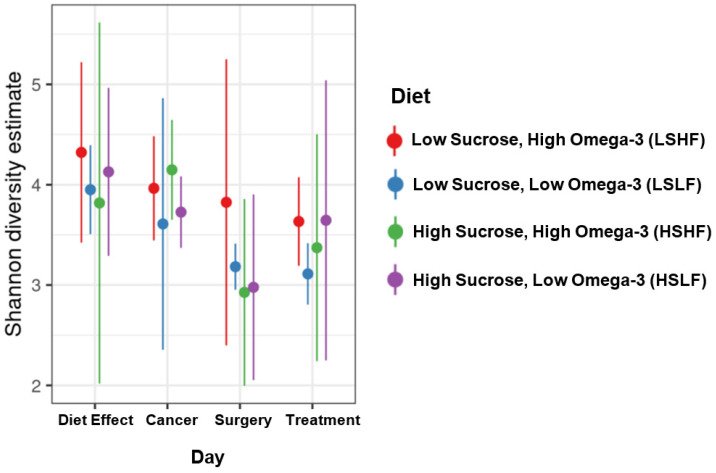
Richness and evenness of ASV/per-sample, weighted by the Shannon diversity index, demonstrate differences in alpha-diversity modeled by day and diet.

**Figure 4 nutrients-15-00724-f004:**
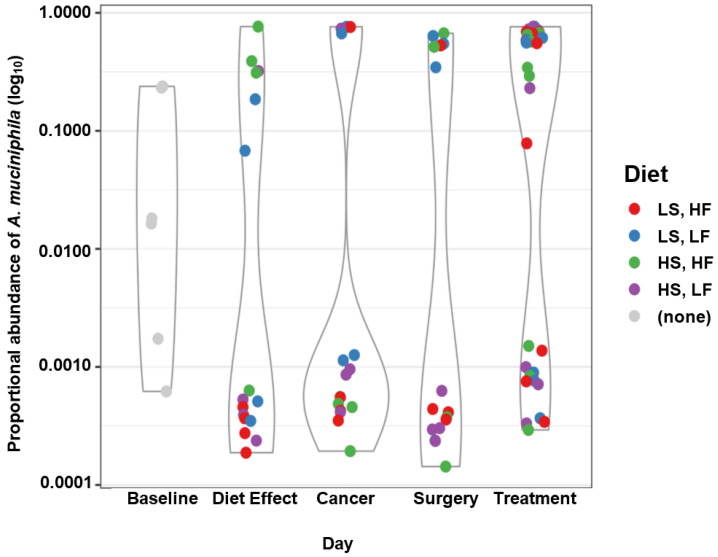
Abundance of *Akkermansia muciniphila* changes over time.

**Figure 5 nutrients-15-00724-f005:**
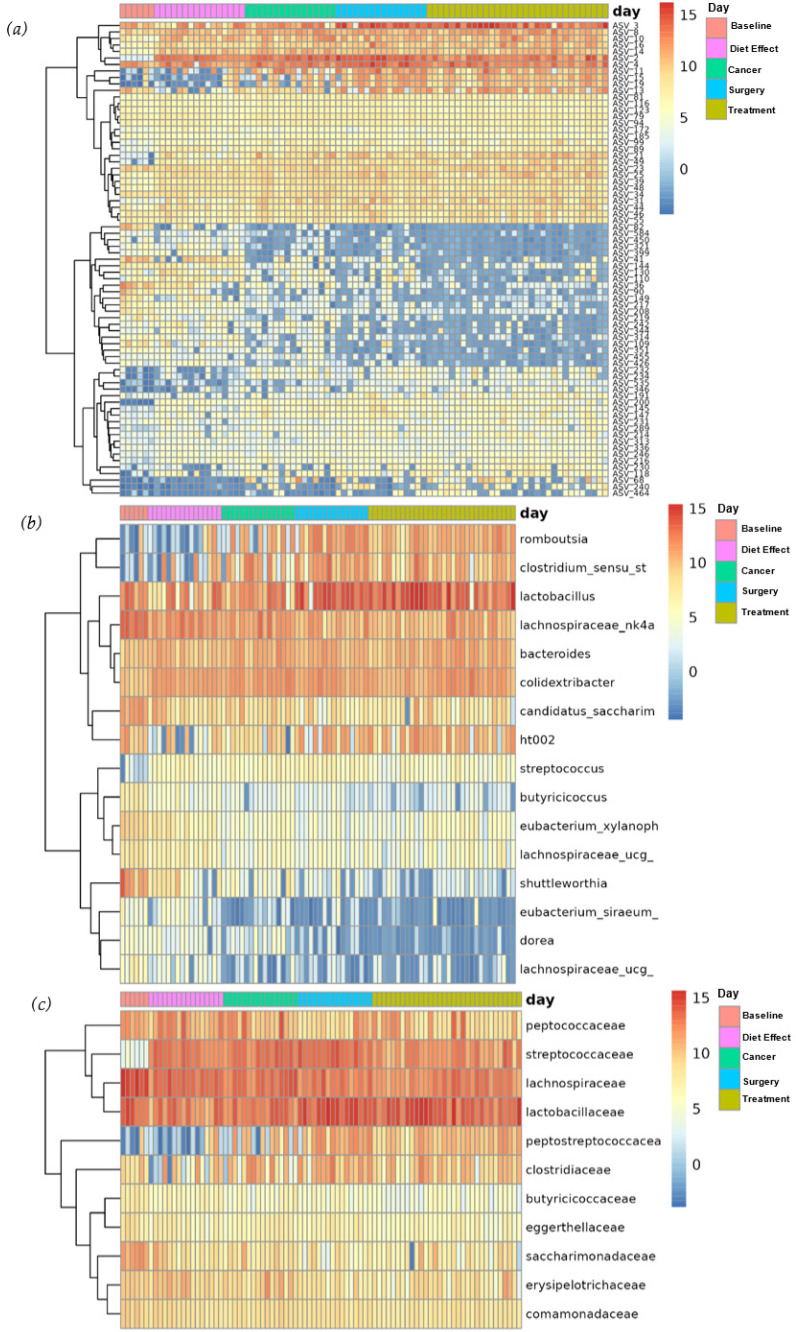
(**a**) Abundance heatmap of ASVs by day; (**b**) abundance heatmap of genera by day; (**c**) abundance heatmap of families by day. Only features with an adjusted *p*-value below 0.1 are shown.

**Figure 6 nutrients-15-00724-f006:**
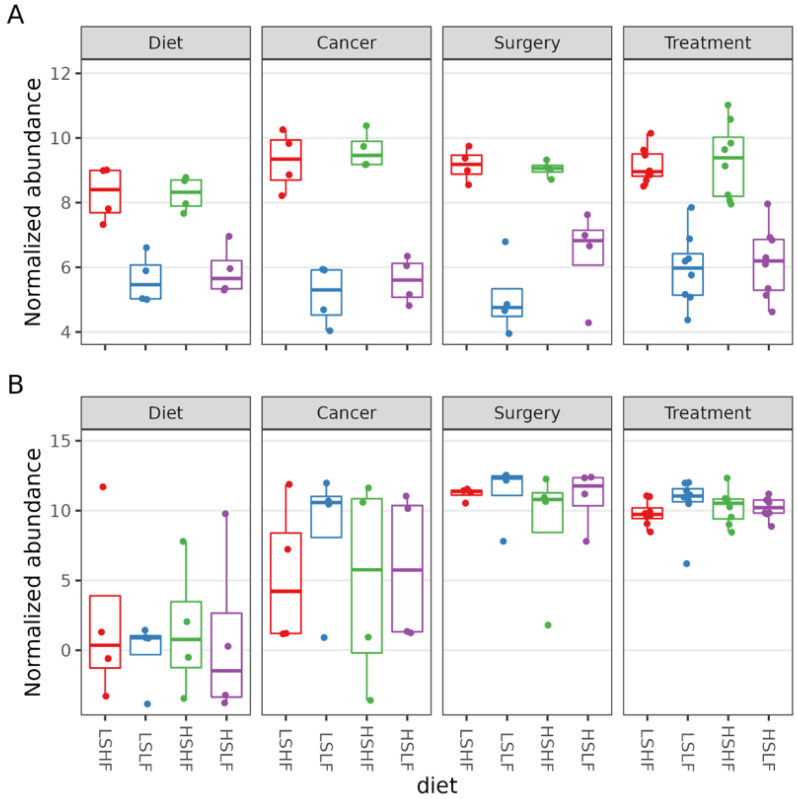
(**A**) Abundance of ASV50 by timepoint and diet; (**B**) abundance of the family *Peptostreptococcaceae* by timepoint and diet.

**Table 1 nutrients-15-00724-t001:** Comparison of richness of ASV/per-sample modeled by day and diet. Number of taxa in observed time points and dietary combinations demonstrate significant effects of day, diet, and their interactions.

Variable	Day	Diet × Day
Diet Effect ^1^	−132.4 (15.7) ***	
Cancer	−164.3 (16.1) ***	−33.4 (15.3) *
Surgery	−219.1 (16.2) ***	−61.5 (15.5) ***
Treatment	−263.3 (11.6) ***	−139.8 (11.1) ***
LS, LF ^2^		−36.3 (14.1) **
HS, HF		−90.5 (13.3) ***
HS, LF		11.6 (13.9)
LS, LF × Cancer		−44.9 (34.3)
HS, HF × Cancer		86.9 (28.4) **
HS, LF × Cancer		−58.9 (29.1) *
LS, LF × Surgery		−59.2 (32.0) +
HS, HF × Surgery		12.8 (32.0)
HS, LF × Surgery		−80.6 (31.4) **
LS, LF × Treatment		−27.8 (22.5)
HS, HF × Treatment		68.4 (21.1) ***
HS, LF × Treatment		−16.9 (22.3)
AIC ^3^	968.5	906.7
Log. Lik.	−478.268	−436.327
formula	estimate ~ day	estimate ~ diet × day

^1^ + *p* < 0.1, * *p* < 0.05, ** *p* < 0.01, *** *p* < 0.001; ^2^ LS, LF: low sucrose, low omega-3 fatty acid diet. LS, HF: low sucrose, high omega 3 fatty acid diet. HS, LF: high sucrose, low omega-3 fatty acid diet. HS, HF: high sucrose, high omega 3 fatty acid diet; ^3^ AIC: Akaike Information Criterion, where lower values indicate a better model fit.

## Data Availability

All code and microbiome population analysis are available online at https://github.com/jelmerp/mouse-cancer-metabarcoding (accessed on 26 January 2023).

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
