# Peer review of "Dietary Impacts on Changes in Diversity and Abundance of the Murine Microbiome during Progression and Treatment of Cancer"

_nutrients, 2023, doi:10.3390/nu15030724_

Round 1
Reviewer 1 Report
This manuscript by Holly Paden et al. describes a study of the microbiome during breast cancer progression and treatment in a murine model using 4 different diets.
The study and the data that was gathered has a high relevance to the development of the microbiome under cancer treatment in the background of different diets, because it might be involved in the treatment outcome.
However, the manuscript needs to be improved especially in the way the data is presented and discussed in order to enable readers to understand the implications. For example, it still remains unclear to me if the different diets were merely meant to create some background variation ("to emulate dietary recommendations and common intake levels in a Western diet", L. 79) or if diet-specific effects were of actual interest. The main reason for this is that the latter are not presented and discussed in full detail.
Given that this manuscript was submitted to "Nutrients", I would actually expect that the impact of different diets is thoroughly assessed and described (although it certainly would be no less interesting to a more cancer-oriented community).
I will try to sum up the major points that I would like to suggest improving:
1) Please clearly describe the effects/differences that occur at each step of the progression (switch to study diet, cancer induction, surgery, treatment), considering in full the other experimental factors diet and treatment. In part, this is already present, but most of the time, some important information is left out. Examples are:
i) differences in weight gain between diets are presented in 3.1. Data on the effect size (how many grams?) is missing. ii) differences in richness are presented as boxplots for each diet (Fig 1) and as effects in a statistical model (Tab 1). The differences between diets in both are not described and never discussed.
iii) A PERMANOVA on beta diversity/microbial composition is presented as different in a model including diet and day. Similarities and differences betwenn individual diets are not described and the supplementary PCoA plot does not include diet as a factor. For the PERMANOVA it would also be helpful to provide R-square values to allow a comparison of the effects of time (=progression phase) and diet.
iv) L.263ff, Figure 5: the text refers to differences between the diets but the figure does not differentiate diets.
2) Not all samples were selected for analysis, probably due to limited resources. This is alright, but the samples to analyze seem to have been selected randomly for each timepoint, creating some issues for the analysis and for statistics. If subjects (here: mice) appear several times in an analysis, the samples are statistically dependent and the mouse ID is an additional confounding factor (might be seen as an individual baseline for each animal). This seems to be ignored here and it might cause some problems with statistics. Ideally, the same mice could have been selected at each timepoint to improve the calculation of the confounding effect, while choosing random subjects at each timepoint creates a number of missing data. Also, interpretation of the bimodality of ASV1 (Figure 3) and of Figure S5 would be easier, if individual trajectories could be tracked for all timepoints across individuals (Figure S5 attempts to do this but most mice seem to have only a selection of datapoints). This is for sure difficult to improve retrospectively but I suggest to at least try to take it into account in the statistical tests.
3) Statistical models need to be described better. E.g., in Table 1, it might confuse readers that diet A is not shown (likely because it acts as intercept in the model). The factor "diet" in the statistical models could easily be misunderstood. If I understand correctly, in most cases presented here it means the effect of shifting diet from chow to one of the semi-purified diets. Given the objectives of the study, one would likely expect "diet" to refer to the effect of the four different diets. Since it refers to the pre-cancer phase, it could be defined as "diet baseline" or similar to prevent confusion. Likewise, the factor "Chemotherapy" is not clear. Does it refer to the timepoint or is a difference between treated/untreated animals considered? In Table 1, it might confuse readers that diet A is not shown (likely because it acts as intercept in the model).
4) Please give detailed information on the composition of the composition of the diets? What were the ingredients? How much energy did they contain per gram? To properly assess weight gain or loss, it is important to know the energy uptake of the animals. Also, if any compononent (e.g. sucrose) is "high", at least one other component needs to be low.
5) Akkermansia was likely removed from the statistics because of its high abundance with bimodal distribution. Was the effect of the removal on the Aldex statistics actually tested? Since Aldex was designed for compositional data, I think it might be problematic to remove a substantial amount of reads before the clr transformation. The dominant effect of Akkermansia on beta diversity in Figure S4a could probably also be mitigated by trying higher PCo dimensions (axis 3, 4, etc...).
6) The discussion part should include more own data. It consists predominantly of context (literature). For example, it would be interesting to compare abundances of mollicutes, which are mentioned in the introduction (L.65) but not consired later on.
MINOR comments
7) chapter 3.1 title lists "tumor growth" but no data is presented
8) diet name usage is inconsistent (A/B/C/D or HS-HF, LS-LF, ...)
9) L. 277: italization of species and genus names is inconsistent (should always be italics)
10) L. 280: why 5 taxa? 4 species were mentioned
11) Table 1: AIC is presented but not interpreted, if it's not used, it could be removed
Reviewer 2 Report
Thank you for the opportunity to revise the manuscript. Here some observations:
Introduction
The introduction overall needs more clarity. Perhaps, the main effect of cancer treatments in the gut it is potentially dysbiosis reflected as an imbalance in the composition and diversity of the gut microbiome but this message it is lost in the introduction and in discussion.
In the first paragraph consider focusing only on breast cancer statistics which is the focus of your study.
Reference 1 and a are the same from different years, perhaps keep the one that is more recent.
Iam not so sure why in the second paragraph you mention susceptibility to infection if you are not looking into that. Your hypothesis mentions inflammation.
You mention several times treatment outcomes but your study does not focus in treatment outcomes. The introduction could benefit from a hypothetical explanation or a link of how diet affects the stool microbiome and inflammation.
The introduction can benefit from a little more information related to dietary recommendations and western diet. You mention in the hypothesis dietary recommendations but there is no mention of diet guidelines/recommendation for cancer patients. There is a lot of literature on Western diet and cancer but only one sentence it related to western diet.
In addition, the introduction could benefit from a point of what is the importance of conducting your study in animal models. Clinical studies has been conducted in that area.
The Objective and hypothesis of the study it is very interesting. However,
1- For clarity, I believe you are looking into the stool microbiome. For gut microbiome perhaps you may look into biopsies of the gut tissue. Or you are using stool as proxy of the gut microbiome.
2- Is the objective of the study to determine changes in the gut microbiome OR changes in diversity and composition of the stool microbiome…
3- Iam a bit confuse with progression of breast cancer, surgery and chemotherapy. All the animals went to all the phases. Did you had control group???
Materials and methods
2.2 Experimental design, Overall the design it is very confusing. It could benefit from a picture/diagram. It is very hard to follow. Did you ever stop diet? Why D6 is diet effect? Why D29 is surgery effect if they already had ovariectomy? Why D29 is treatment effect? All mice had diet + cancer + surgery + chemo? Did all the mice had diet? If you do not have a control group how can you determine one specific effect?
Is the postmenopausal state common to the majority of breast cancer women because they undergoes chemotherapy???
Why introducing semi-purified diet please explain, is that part of the intervention or not???
Ten days following lumpectomy, mice from each dietary cohort were treated with either a saline ... . BUT YOU HAVE NOT MENTIONED/DETAILS ABOUT DIETARY COHORTS
2.3 Diet
Which diet is simulating Western diet? Can you clarify what each diet represent.
Seems that Diet B is plant-based but you do not mention plant-based in the introduction.
2.5 First time you mention treatment combination
2.6 Morphological Analysis
Needs clarification. What is the outcome in this section?
3. Results
Consider informing your results according to diet A,B,C,D. You have 2 low sucrose and 2 high sucrose
Same for Table 1 in the text it is diet A,B,C,D and in table is LS, LF2 x cancer. How come you have 4 diets and in the table seems to be only 3 diets per each cycle of treatment.
3.2
This is like a hanging sentence: Other alpha-diversity measures, such as the Shannon index, additionally take even-235 ness in abundance into account. I am not so sure why it is in results.
3.3 Please, clarify. How can you identify species from 16S? Or are you predicting species. Why A. municiplila in Figure 3? Did you identify more species? Only A. muciniphila was important? Then, Figure 4 are heat plot with genus levels ? Why staring with a species and go back to genera? If you can identify species, then reporting genera it is not important.
3.5
Iam not so sure why reporting one paragraph on genera, one paragraph on species, one paragraph on families and one paragraph on order level. Again, if you for sure can identify species from your analisis, then I am not so sure why reporting families and order. Were does families and order different to species or genera?
4 Discussion
The discussion it is interesting.
Paragraph 1, Can you clarify and link your findings to the discussion. The changes seen here can largely be attributed to a decrease in overall richness throughout disease progression. What changes are you referring to?? The potential physiological explanation for the changes in the microbiome it is not discussed. Breast cancer not discussed.
What is a bimodal polarization and why it is important?
This is a nutrition journal. Having more discussion on nutrition will be of interest to the reader of the journal. Can you discuss Western diet etc.?
Conclusion.
It is the first time you mention macronutrient dietary composition??? Can you link your results to the conclusion. Your study it is in animal models, the conclusion sounds like your study it is in humans. Perhaps focus on how your findings can be translated to clinical studies
Round 2
Reviewer 1 Report
The authors have adressed all my specific suggestions and thereby improved the presentation and make it a lot easier for the reader to understand the paper.
There is still some room for improvement, which I haven't adressed specifically in my first review, or which only now occured in the second version.
Before I come to specific suggestions I would like to repeat my statement, that it is difficult for the reader to identify the most important or relevant observations. Besides my specific suggestions in the first review, I already expressed this and despite the improvements it still seems not fully clear to me. What are the main points? Is it the description of the microbiome during disease progression and treatment? Or the effect of the diets? Or the effect of the treatment?
My interpretation is that this particular work might be part of another study, which probably had somewhat different aims. This would explain why only some of the animals were actually used for the microbiome analysis. If this is the case, I'd suggest to make this point clear to the reader as it would really help to prevent some confusion. And it would also explain why the cohorts don't seem to fully fit with the way the observations are described and interpreted. I am not sure if I am able to make myself clear at this point. I feel like the text is somewhat inconsistent in deciding which parts of the design to emphasize and which parts are only mentioned as a sort of side note. Maybe this becomes more clear with the point of the treated vs untreated mice (the last of the five timepoints where samples were taken). Initially the authors speak of eight cohorts (4 diets plus treated/untreated) but most of the graphs don't differentiate between treated/untreated. Only the supplement and a few sentences in the text are spent on this feature of the experimental design. If this was designed on purpose and not as a part of a related but different research aim (which necessarily doubled the amount of animals needed for the study!), I'd expect more discussion of this aspect.
I understand, this is a very general point and not all readers might have the same difficulties with understanding the purpose of the text that I experienced. Still, I believe that this could be improved and would like to add some further suggestions:
1) The hypotheses need to be stated clearly in the introduction and should fit the experimental design. While two hypotheses are stated (L. 83-87), there is no mention of the effect of chemotherapy itself, which accounts for doubling the amount of animals used for the study (8 instead of 4 cohorts).
2) The hypotheses should be reflected in the discussion and conclusions. While aspects of both proposed hypotheses are somewhat discussed, it could be presented much more clearly if the data confirm or contradict the hypotheses. Especially the question of pathogens and inflammation seems a bit scattered in the discussion. It could help to add some summarizing sentences. And the conclusions are currently very general and contain little input of the actual results. Reflecting the hypotheses more clearly in this section could also help here.
3) The hypothesis that diversity and richness decrease over the cancer and treatment progression seems to be confirmed by the data. However, there is an important limitation by design. Without an untreated control group, all the observation could also be explained as experimental side effects, e.g. caused by the prolonged feeding with a semipurified diet (usually devoid of complex fibre content that would promote diversity). This limitation should be discussed.
4) Figure 2 suggests that for the most part of the study there were not 8 but only 4 cohorts, since all boxplots expect "Treatment" include only 4 data points. If the authors don't want to emphasize on the treatment anyway (see above), I would suggest to only mention 4 cohorts and instead explain that the last timepoint was split in two groups per cohort, leading to 8 instead of 4 data points. I also believe that "Treatment" might still confuse readers, expecially if the data points actually refer to treated AND untreated mice. I suggest splitting this timepoint and display the two subgroups independenty. Thereby, differences (or absence thereof) between the groups would be visible. This could also be applied to most of the other figures.
5) Figure 1 was a useful addition. If a) and b) were merged, the differences between day 1 and necropsy would be easier to spot. How does "necropsy" relate to the other timepoints? Is it identical with "Treatment"?
6) In section 3.4, the LF/HF diets are referred to as "low fat" or "high fat", which is incorrect, since the macronutrients (fat in general) should be identical. I understood that only the type of fat differed.
7) Akkermansia is apparently polarized. The individual mice data suggests that the abundance switches between high and low state even for individuals and in both directions over the course of the experiment. Responder/Non-responder (L. 353) doesn't seem like a plausible explanation to me, because this would lead to mmice with either high or low abundance. An alternative explanation would be a bistable abundance distribution that switches within each mouse due to unknown reasons (one might also say "randomly").
8) L. 13 "evenness of over", delete "of"
9) L. 260: This sentence seems overly complicated (or wrong).
